# Research

ecology, evolution

*Arabidopsis halleri*, clonal sharing, foraging, herbivore defence, induced defence, metal hyperaccumulation

**Author for correspondence:**
Michal Gruntman
e-mail: mgruntman@tauex.tau.ac.il

# Between the devil and the deep blue sea: herbivory induces foraging for and uptake of cadmium in a metal hyperaccumulating plant

Anubhav Mohiley[1], Tanja Laaser[1], Stephan Höreth[3], Stephan Clemens[3], Katja Tielbörger[1,†] and Michal Gruntman[1,2,†]

[1]Plant Ecology Group, Institute for Evolution and Ecology, University of Tübingen, Tübingen, Germany
[2]School of Plant Sciences and Food Security and Porter School of the Environment and Earth Sciences, Tel Aviv University, Tel Aviv, Israel
[3]Lehrstuhl Pflanzenphysiologie, Universität Bayreuth, Bayreuth

AM, 0000-0002-7690-5231; MG, 0000-0003-2228-9266

Plants have been shown to change their foraging behaviour in response to resource heterogeneity. However, an unexplored hypothesis is that foraging could be induced by environmental stressors, such as herbivory, which might increase the demand for particular resources, such as those required for herbivore defence. This study examined the way simulated herbivory affects both root foraging for and uptake of cadmium (Cd), in the metal-hyperaccumulating plant *Arabidopsis halleri*, which uses this heavy metal as herbivore defence. Simulated herbivory elicited enhanced relative allocation of roots to Cd-rich patches as well as enhanced Cd uptake, and these responses were exhibited particularly by plants from non-metalliferous origin, which have lower metal tolerance. By contrast, plants from a metalliferous origin, which are more tolerant to Cd, did not show any preference in root allocation, yet enhanced Cd sharing between ramets when exposed to herbivory. These results suggest that foraging for heavy metals, as well as their uptake and clonal-sharing, could be stimulated in *A. halleri* by herbivory impact. Our study provides first support for the idea that herbivory can induce not only defence responses in plants but also affect their foraging, resource uptake and clonal sharing responses.

## 1. Introduction

In natural ecosystems, plants experience spatial and temporal heterogeneity of resources. In response to such resource heterogeneity, plants have been shown to adjust their foraging behaviour and selectively place and proliferate their resource-acquiring organs within resource-rich patches [1–5]. The two most studied types of such foraging behaviour in plants are root-foraging patterns displayed in response to patchy distribution of soil nutrients [3–5]), and foraging by clonal plants, which exhibit active placement of daughter ramets in rich patches [1,6,7]. In addition, clonal plants may maximize their performance by division of labour among ramets that grow in patches of varying resource availability and by sharing of different resources taken up by individual ramets [8,9].

Many studies have shown that foraging in plants can be elicited by resource heterogeneity [3,6,10]. Other studies have also indicated that these foraging decisions can be affected by the temporal variance of resources [10] or the presence of competitors [11]. However, despite the overwhelming importance of

†These authors contributed equally.

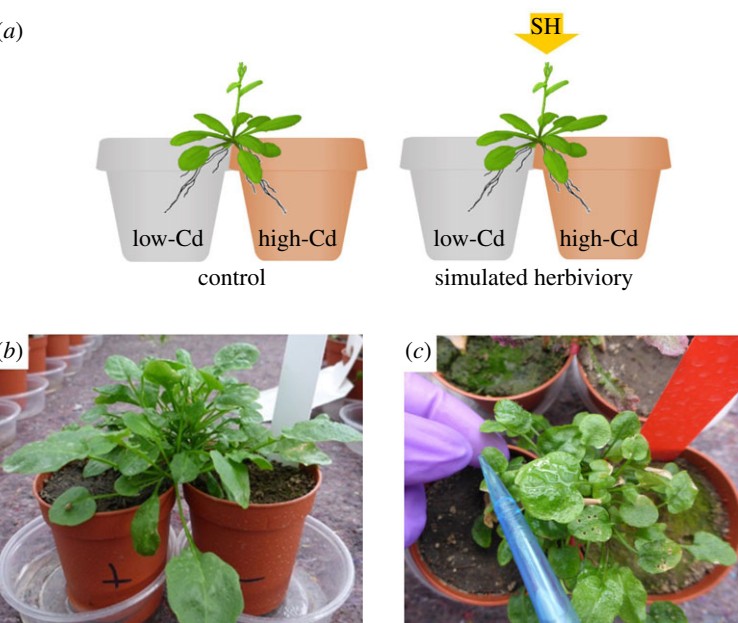

**Figure 1.** Schematic illustration of the root-foraging experiment (*a*) with a picture depicting an *A. halleri* ramet growing in a split-root set-up in paired pots (*b*) and a picture depicting the simulated herbivory treatment (SH) with leaf piercing and jasmonic acid application (*c*). (Online version in colour.)

biotic interactions in general, and enemies in particular, for determining demand and supply of resources for plants, the role of enemies in inducing and modulating foraging decisions in plants has been seldom explored. Specifically, herbivory is known to induce varying defence-related physiological and morphological responses in plants, such as the production of secondary metabolites due to both foliar and root herbivory [12–14]. Herbivore damage might therefore increase the demand for certain resources that are required for the production of such resistance compounds and hence affect the foraging decisions of plants. However, to the best of our knowledge, this hypothesis has not been tested to date.

So far, foraging decisions in plants have been most commonly studied with respect to resources such as light, water or nutrients, which are required for growth, reproduction or maintenance of physiological processes. A seldom explored idea is that under certain conditions, plants might also forage for substances that are detrimental to them and decrease their fitness. A number of such plant species can be found in the Brassicaceae family that hyperaccumulate heavy metals such as zinc (Zn), cadmium (Cd) or nickel (Ni) up to 100–1000-fold higher than those found in non-hyperaccumulating species [15]. Several hypotheses have been suggested to explain why such behaviour could be beneficial, and the most common is the elemental defence hypothesis, which suggests that heavy metals could serve as herbivore defence [16–19]. Interestingly, some metal-hyperaccumulating plants have been shown to forage for heavy metals [20,21]. For example, Dechamps *et al.* [21] showed that the metal hyperaccumulator *Noccaea caerulescens* allocated more root in high metal patches, in response to heterogeneity in metals. If this is the case, then metal foraging and uptake in these plants might be further enhanced by herbivory as an induced defence mechanism.

In this study, we present two independent experiments that examined the hypothesis that foraging for, uptake and sharing of heavy metals can be induced by herbivory. Specifically, we studied these responses in the metal-hyperaccumulating clonal plant *Arabidopsis halleri*. This species can accumulate large concentrations of heavy metals (Zn and

Cd) in its shoots and leaves [16–18], and these have been shown to deter herbivores [16,17,19]. In a first experiment, we asked whether simulated herbivore damage induces root foraging for Cd within ramets by studying root allocation in a 'split-root' design between Cd-rich and Cd-poor patches (figure 1). In a second experiment, we asked if simulated damage induces both increased Cd uptake as well as increased Cd sharing between *A. halleri* ramets (figure 2). Here, we also asked whether there is a difference in Cd uptake and sharing when herbivory is induced in ramets growing in a Cd-rich patch or in connected ramets growing in a Cd-poor patch (figure 2). In both experiments, we additionally differentiated between Cd-tolerant plants and plants for which Cd is more harmful in order to evaluate differences in their foraging decisions. Specifically, we asked if responses to herbivory differ between plants from metalliferous versus non-metalliferous origin. *A. halleri* from both these origins have been shown to hyperaccumulate Cd [19,22,23]. However, results from a previous study with the same genotypes used in this study showed that *A. halleri* from a non-metalliferous origin are less tolerant to high concentrations of Cd in their tissues and showed markedly reduced growth when grown in Cd-rich soils, while plants from metalliferous soil were not affected [22].

## 2. Material and methods

### (a) Plant and soil

*A. halleri* individuals for both experiments were collected in December 2013 from four metalliferous sites (i.e. abandoned mining areas) and four non-metalliferous sites within Germany (table 1). These individuals were also used in a previous experiment that showed low-Cd tolerance of plants from a non-metalliferous origin [22]. Twenty individuals were collected per site in an arbitrary manner. However, we applied some stratification and ensured a minimum distance of 2 m and a maximum distance of 150 m between individuals, to make sure they belonged to different genets. In December 2013, collected individuals were

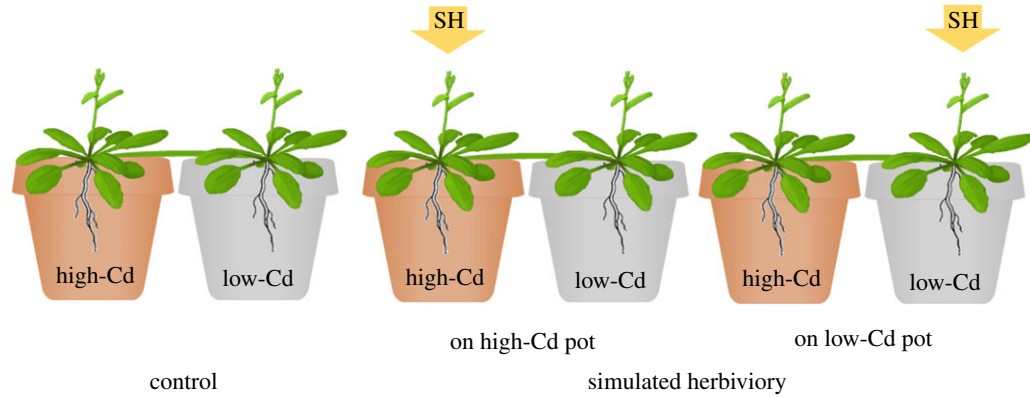

**Figure 2.** Schematic illustration of the clonal-sharing experiment with connected *A. halleri* ramets growing in separate pots and subjected to simulated herbivory (SH) treatments with leaf piercing and jasmonic acid application. (Online version in colour.)

**Table 1.** Information about source populations of *A. halleri* used in the root-foraging experiment and clonal-foraging experiment.

| origin | population | latitude | longitude |
|---|---|---|---|
| non-metalliferous | Blaibach[a,b] | 49°09.830 N | 012°47.759 E |
| | Fort Fun[a,b] | 51°18.264 N | 010°18.004 E |
| | Geroldsgrün[a,b] | 50°23.323 N | 011°34.148 E |
| | Wehbach[a,b] | 50°48.498 N | 007°50.563 E |
| metalliferous | Clausthal Zellerfeld[a,b] | 51°48.088 N | 010°18.111 E |
| | Lautenthal[b] | 51°51.453 N | 010°18.004 E |
| | Littfeld[a] | 51°00.540 N | 008°00.660 E |
| | Vienenburg[a,b] | 51°57.294 N | 010°34.082 E |
| | Wulmeringshauen[a,b] | 51°18.383 N | 008°29.112 E |

[a]Root-foraging experiment.
[b]Clonal-foraging experiment.

planted in 1 l pots filled with potting soil (Topferde, Einheitserde, Gebr. Patzer GmbH & Co. KG, Kreutztal, Germany) and placed in a greenhouse in Tübingen University, Germany. In order to avoid maternal effects due to metal remains in plant tissues, the plants were clonally propagated for two generations until the beginning of each experiment for which new cuttings were obtained from the propagated clones.

The soil used in the first experiment (root foraging) was collected from the same metalliferous and non-metalliferous sites where *A. halleri* was sampled (table 1) and at the same time. The soil was collected at a depth of 30 cm from three locations within each site. In order to minimize potential differences between the soils in their physical properties, nutrient availability and the presence of soil mutualists or antagonists, the soils were sieved (2 mm mesh size) and steam-sterilized for 2.5 h at 80°C and mixed with 10 g of slow-release fertilizer (Osmocote Classic 14% N, 14% $P_2O_5$, 14% $K_2O$; Scotts, Geldermal-sen, The Netherlands). All soils from the same type (metalliferous versus non-metalliferous) were then mixed to avoid any confounding effects of local adaptation to home soil. Cd concentration was markedly greater for metalliferous soils compared to non-metalliferous soils (29.04 versus 4.71 µg g$^{-1}$ dry soil; see below for explanations of analyses). The pH of the metalliferous and non-metalliferous soil mixtures was 5.4 and 5.8, respectively.

For the second experiment (clonal sharing), we were not able to obtain additional field soil and therefore chose to use similar non-contaminated local field soil (Bischoff GmbH & Co. KG, Hirschau, Germany), which we artificially contaminated. The soil was sieved (2 mm mesh size) and autoclaved for 20 min at 120°C and half of it was artificially contaminated with 100 ppm Cd by adding $CdCl_2$ (99%, Sigma-Aldrich Chemie GmbH, Germany) solution to the soil.

## (b) Root-foraging experiment

The experiment took place in a greenhouse at Tübingen under natural light conditions and with temperatures between 20 and 35°C. In May 2014, newly grown ramets of *A. halleri* were selected and severed from eight randomly selected mother plants per population. Each ramet was grown in water-filled containers in the greenhouse to induce root formation [24]. After two weeks, the ramets produced 6–10 roots, out of which all except two similarly sized roots where severed. Each ramet was then grown in a split-root set-up of paired 0.05 l pots with one root in a low-Cd pot (with soil from non-metalliferous sites) and the other in a high-Cd pot (with soil from metalliferous sites) (figure 1*a,b*). Ramets were then assigned to either a control (no-herbivory) treatment or a simulated herbivory treatment. Herbivory was simulated by mechanical damage, through puncturing holes in the leaves, combined with jasmonic acid (JA) application (figure 1*c*) [25]. One millimolar JA was used and the solution was prepared by mixing 250 mg of JA (Sigma-Aldrich Chemie GmbH, Germany) with 1 ml of ethanol and 250 ml of demineralized water, after which 2.5 ml Triton X-100 (0.1%) were added [25]. Three hundred microlitres of the solution was applied using a pipette on one leaf per ramet after piercing six holes in it using a toothpick (figure 1*c*). This procedure was repeated once every 7 days, until two weeks before harvesting. In total, the herbivory treatment was applied four times during the experiment. Ramets in the no-herbivore control treatment were applied with a solution without JA but with 2.5 ml HCl to obtain the same pH [25]. This method has been previously used to simulate herbivory, as the combined application of damage and JA covers the full response spectrum to herbivory [25]. Water was provided as per the requirement of the plant, which was approximately twice a week. The experimental set-up consisted of 128 pot pairs [2 herbivory treatments × 2 plant origins (metalliferous, non-metalliferous) × 4 populations × 8 individuals]. However, 27 of the ramets died during the experiment and were therefore excluded from the analyses.

The plants were harvested after six weeks, following which the shoot biomass was harvested, roots were washed and their biomass was measured after oven drying for 3 days at 60°C. However, shoot biomass of six plants per treatment could not be analysed, as they were used for additional chemical analysis (data not shown).

## (c) Clonal sharing experiment

In April 2015, connected ramet pairs of *A. halleri* with a stolon length of 2.5–4 cm were selected and cut off from each of the

**Table 2.** Results of the root foraging experiment. GLMMs were used to investigate the effects of simulated herbivory (control versus simulated herbivory) and *A. halleri* origin (metalliferous versus non-metalliferous) on shoot biomass, and the effects of simulated herbivory, *A. halleri* origin and pot (low versus high Cd) on root biomass of *A. halleri*. Population and genotype nested within population were used as random factors. Significant values are indicated in italics. *F* is for the fixed effects and Wald *Z* for the random factors.

| fixed factors | shoot biomass (mg) | | | root biomass (mg) | | |
|---|---|---|---|---|---|---|
| | d.f. | *F* | *p* | d.f. | *F* | *p* |
| simulated herbivory (H) | 1 | 2.152 | 0.146 | 1 | 4.6 | *0.033* |
| origin (O) | 1 | *10.241* | *0.002* | *1* | *17.692* | *0.001* |
| pot (P) | | | | 1 | 2.463 | 0.188 |
| H × O | 1 | 0.011 | 0.918 | 1 | 0.51 | 0.476 |
| H × P | | | | 1 | 0.112 | 0.739 |
| O × P | | | | 1 | 8.591 | *0.004* |
| H × O × P | | | | 1 | 4.36 | *0.038* |
| **variance** | **d.f.** | **Wald Z** | ***p*** | **d.f.** | **Wald Z** | ***p*** |
| population | 7 | *6.038* | *0.001* | 7 | *9.718* | *0.001* |
| genotype (population) | 7 | *2.389* | *0.017* | 7 | *2.58* | *0.01* |

same eight mother plants per population. Ramet pairs were grown in paired 0.05 l pots with one ramet in a low-Cd (non-contaminated) pot and the other in a high-Cd pot (100 ppm Cd), (figure 2). One month after the beginning of the experiment, when leaves reached a length of 2 cm, the paired ramets were randomly assigned to a control (no-herbivory) treatment or one of two simulated herbivory treatments, which were applied on the ramet in either the high-Cd or low-Cd pot (figure 2). The simulated herbivory was applied as in the root-foraging experiment, except that the total herbivory application was six times. The experimental set-up consisted of 192 pot pairs [3 herbivory treatments × 2 plant origins (metalliferous, non-metalliferous) × 4 populations × 8 individuals]. However, during the experiment, 30 ramet pairs died and in 14 others, one ramet died and these ramet pairs were therefore excluded from the analyses.

As the resource flow between ramets might be unidirectional from older to younger ramets [26], the position of the two paired ramets was alternated between replicates so that in half of the pairs the mother ramets were assigned to the high-Cd pot, while in the other half, the daughter ramets were assigned to it. The paired pots were placed in the greenhouse. Each pot was placed within a separate plastic dish (6 mm) to allow for their individual watering. Water was provided approximately twice a week. The plants were harvested after four months in August 2015.

We chose to study the uptake of Cd in this experiment as previous studies have shown that even though both Cd and Zn accumulation by *A. halleri* can act as herbivore defence, Cd has a much greater potency as a defence compound and requires smaller quantities to be effective [16]. Hence, leaves from the experiment were analysed for Cd concentration. Leaf extracts were prepared with the same methodology as in our own previous experiment [22] and analysed with the ICP-OES (iCAP 6500, Thermo Scientific) technique for Cd quantification [23].

### (d) Data analysis

In the root-foraging experiment, a generalized linear mixed model (GLMM) was used to examine the effect of simulated herbivory and *A. halleri* origin (metalliferous versus non-metalliferous) on the foraging decisions of *A. halleri* between high versus low-Cd pots, with root biomass as the response variable, and herbivory, origin, pot type (high versus low-Cd) and their interactions as fixed factors, and genotype nested within population and population as random factors. Similarly, effects on shoot biomass were analysed using a GLMM, with herbivory, origin and their interactions as fixed factors, and genotype nested within population and population as random factors. Effects on root biomass were analysed using a gamma probability distribution with an identity link function, while effects on shoot biomass were analysed using a normal probability distribution with a log link function.

In the clonal-sharing experiment, a GLMM was used to examine the effect of herbivory, origin and pot type on shoot biomass and Cd accumulation of *A. halleri*, with accumulated leaf Cd concentration as the response variables and herbivory, origin, pot type and their interactions as fixed factors, and genotype nested within population and population as random factors. We initially added ramet identity (mother or daughter) to the analysis, to learn if it might affect Cd allocation patterns, but removed it due to lower model fit (a higher AIC value) and lack of statistically significant effect. Shoot biomass was analysed using a normal probability distribution with a log link function, while leaf accumulated Cd concentration within ramet pairs was analysed using a gamma probability distribution with a log link function. For all analyses, differences between treatments were analysed using post hoc pairwise comparisons with the false discovery rate correction [27]. IBM SPSS Statistics 22 was used for all the statistical analyses.

## 3. Results

In the root-foraging experiment, *A. halleri* from non-metalliferous origin had higher shoot biomass compared to plants from metalliferous origin (table 2, origin effect; figure 3a). However, shoot biomass was not affected by simulated herbivory (table 2, herbivory effect; figure 3a). *A. halleri* from non-metalliferous origin also produced higher root biomass (table 2, origin effect; figure 3b), but exhibited greater root allocation towards low compared to high-Cd pots, while plants from metalliferous origin exhibited no preference in their root allocation (table 2, origin × pot effect; figure 3b). Furthermore, the greater root allocation exhibited by plants

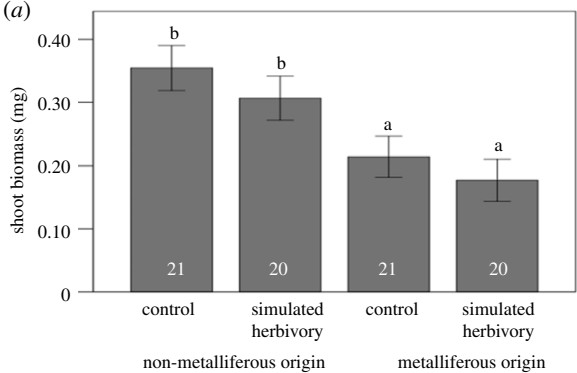

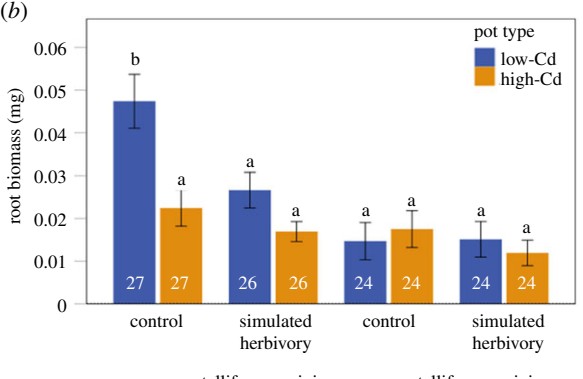

**Figure 3.** Results of the root foraging experiment depicting responses (means ± s.e.) of *A. halleri* from non-metalliferous and metalliferous origin to simulated herbivory and low versus high-Cd pots in (*a*) shoot and (*b*) root biomass. Different letters indicate statistically significant pairwise comparisons (pairwise LSD tests with the false discovery rate correction, [27]). Sample sizes per treatment are indicated in white. (Online version in colour.)

from non-metalliferous origin towards the low-Cd pots was mostly observed under control conditions but diminished under simulated herbivory, while root allocation in high-Cd pots remained the same (table 2, herbivory × origin × pot effect; figure 3*b*).

In the clonal-sharing experiment, shoot biomass of ramets growing in the low versus high-Cd pots differed between *A. halleri* plants from the two origins and in response to simulated herbivory (origin × pot effect, herbivory × pot effect, table 3 and figure 4*a*). Particularly, in pairs from non-metalliferous origin, ramets had higher shoot biomass in the low-Cd pot, except when simulated herbivory was applied on the low-Cd pot, where ramets had similar biomass in both pots (table 3 and figure 4*a*). However, pairs from the metalliferous origin did not show differences in ramet biomass between the high and low-Cd pots, regardless of the herbivory treatment (figure 4*a*).

Cd accumulation was also affected by plant origin and herbivory treatments (origin × pot effect, herbivory × origin effect, table 3 and figure 4*b*). Here, ramet pairs from non-metalliferous origin exhibited increased Cd accumulation under the two simulated herbivory treatments compared to control conditions, whereas pairs from metalliferous origin showed high Cd concentration irrespective of the herbivory treatment (table 3 and figure 4*b*). Moreover, *A. halleri* from non-metalliferous origin exhibited high Cd sharing between ramets found in the low and high-Cd pots, while plants from metalliferous origin restricted Cd allocation to ramets in the high-Cd pots (table 3 and figure 4*b*), but increased Cd sharing

when herbivory was simulated, and in particular when simulated on the low-Cd pot (figure 4*b*).

# 4. Discussion

Our study provides support for the idea that foraging and resource uptake in plants can be induced by herbivory. Intriguingly, *A. halleri* plants did not forage for a 'positive' resource that enhances their growth but for a substance whose uptake would, without herbivory, be avoided. Specifically, plants from non-metalliferous origin, which were shown to be sensitive to Cd [22], maintained root allocation in high-Cd pots and enhanced uptake of Cd when exposed to herbivory, but decreased root allocation and suppressed Cd uptake under control conditions. Moreover, plants from metalliferous origin, which are more tolerant to Cd, enhanced sharing of Cd between ramets when exposed to herbivory, and in particular when herbivory was simulated on ramets growing in low-Cd soil.

Induced responses to herbivory have been shown in a variety of resistance and tolerance traits [18,28–30]. Recently, aphid infection has been shown to be associated with higher concentrations of heavy metals in the phloem of *A. halleri* leaves [18]. Interestingly, the transcription of metal homeostasis genes has been shown to increase in *A. halleri* in response to leaf wounding [31]. Furthermore, a few studies have demonstrated foraging for heavy metals in metal-hyperaccumulating plants [20,21,32]. However, to the best of our knowledge, none of the previous studies has shown that foraging responses in plants for heavy metals or other substances can be induced by herbivory and that this induction is limited to plants that have a lower tolerance to these substances.

The fact that enhanced foraging for and uptake of Cd under simulated herbivory was shown mainly in ramets from non-metalliferous origin might imply that in these plants Cd serves as induced defence, while in ramets from metalliferous origin, which accumulated excessive amounts of Cd regardless of the herbivory treatment, it rather serves as a constitutive defence. The potential use of Cd as induced herbivore defence suggests that plants from non-metalliferous origin might incur a direct toxicity cost of Cd accumulation. Indeed, previous studies (including our own with the same genotypes) have shown that *A. halleri* originating from non-metalliferous populations are less tolerant to Cd [22,33,34], suggesting that in this study, they accumulated this harmful compound only when its benefits might have outweighed its costs. Moreover, Stein *et al.* [23] have shown that *A. halleri* can accumulate Cd to high levels even when growing in soils with very low Cd content, supporting the potential use of Cd as a resistance compound even in non-metalliferous soils. Similar to the results of our study, a study with *N. caerulescens* has shown that nickel accessions actively foraged for Ni, while non-nickel accession showed avoidance strategies by reducing roots in Ni-rich pots [35].

Interestingly, when simulated herbivory enhanced Cd uptake in plants from non-metalliferous origin, it was equally shared between the affected and unaffected ramets. This result implies that in these plants, herbivory also induces the systemic protection of adjacent ramets, which could be easily infected by insect herbivores [36]. By contrast, plants from metalliferous origin, in which Cd uptake was high and unaffected by the herbivory treatment, enhanced Cd sharing between ramets when exposed to herbivory, particularly

**Table 3.** Results of the clonal sharing experiment. GLMMs were used to investigate the effects of simulated herbivory (control versus simulated herbivory), *A. halleri* origin (metalliferous versus non-metalliferous origins) and pot (low versus high Cd) on shoot biomass and Cd accumulation in *A. halleri* leaves. Population and genotype nested within population was used as random factors. Significant values are indicated in italics. *F* is for the fixed effects and Wald *Z* for the random factors.

| fixed factors | shoot biomass (mg) | | | Cd accumulation in leaves (ppm) | | |
|---|---|---|---|---|---|---|
| | d.f. | F | p | d.f. | F | p |
| simulated herbivory (H) | 2 | 0.198 | 0.820 | 2 | *4.547* | *0.012* |
| origin (O) | 1 | 0.099 | 0.101 | 1 | 0.554 | 0.457 |
| pot (P) | 1 | 1.480 | 0.820 | 1 | 3.630 | 0.058 |
| H × O | 2 | 1.743 | 0.177 | 2 | *4.937* | *0.008* |
| H × P | 2 | *3.736* | *0.025* | 2 | 1.881 | 0.155 |
| O × P | 1 | *4.575* | *0.040* | 1 | *6.019* | *0.015* |
| H × O × P | 2 | 0.551 | 0.577 | 2 | 1.475 | 0.231 |
| variance | d.f. | Wald Z | p | d.f. | Wald Z | p |
| population | 7 | 0.626 | 0.532 | 7 | *13.515* | *0.001* |
| genotype (population) | 7 | *2.180* | *0.029* | 7 | *11.010* | *0.001* |

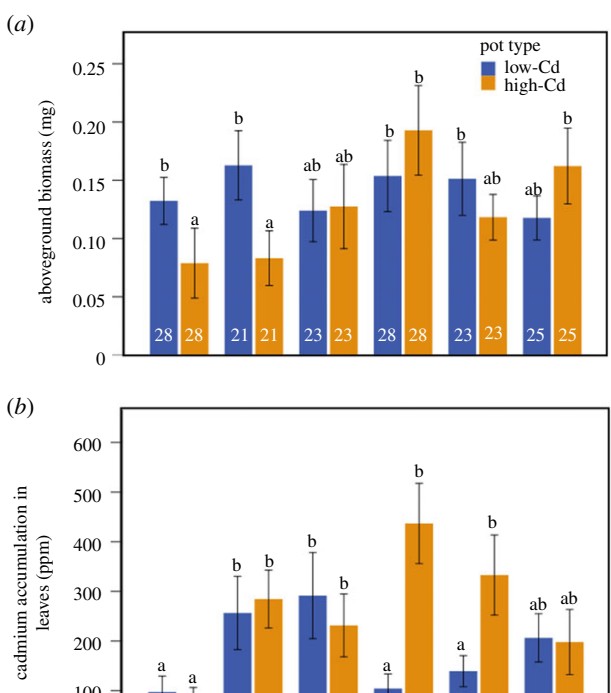

**Figure 4.** Results of the clonal sharing experiment depicting responses (means ± s.e.) of *A. halleri* ramets from non-metalliferous and metalliferous origin to high and low-Cd pots and simulated herbivory (SH) on the high or low-Cd pots, in (*a*) shoot biomass and (*b*) Cd accumulation in the leaves. Different letters indicate statistically significant pairwise comparisons (pairwise LSD test with the false discovery rate correction, [27]). Sample sizes per treatment are indicated in white. (Online version in colour.)

when simulated on ramets growing in low-Cd soil. This result might imply that in these plants, which grow in soils where Cd, and hence herbivore protection, is readily available, Cd relocation is beneficial only to unprotected ramets under attack.

In this study, we used field soil for the root-foraging experiment and artificially contaminated soil for the clonal-sharing experiment, which might differ in the extent of Cd available for the plants, and hence might have affected the root foraging and Cd uptake exhibited in this study. Future studies are therefore needed to determine the role of different soil parameters and artificial metal amendment on the responses of metal hyperaccumulators to herbivory.

## 5. Conclusion

The idea of plant foraging is not new but was studied mainly in relation to the uptake of 'beneficial' substances [3–5,37], and the same applies to studies about sharing of resources in clonal plants [9,38]. Similarly, the idea that herbivore defence could be induced has also been shown previously [18,39]. However, our study is the first to merge these three concepts and demonstrate that foraging and sharing per se is inducible, and that, even more interestingly, foraging and sharing happened for a compound that has an attested negative effect on plant performance. Taken together, our findings demonstrate that foraging for harmful substances, such as heavy metals, can be stimulated when their benefits of protection from herbivory outweigh their costs. This also implies that plants can integrate between two or more very different external signals such as soil Cd (leading to its avoidance) and herbivore attacks (leading to its increased uptake). These results offer insight into the foraging decisions of plants, revealing their ability to integrate complex information on both resource heterogeneity and other environmental stressors [10,11,40].

Data accessibility. Analyses reported in this article can be reproduced using the data provided by Mohiley *et al*. [41].

Authors' contributions. A.M. undertook data curation, methodology and writing-original draft; T.L. undertook data curation, methodology and writing-review and editing; S.H. undertook data curation and writing-review and editing; S.C. undertook data curation and writing-review and editing; K.T. undertook funding acquisition, supervision and writing-review and editing; M.G. undertook funding

acquisition, supervision and writing-original draft. All authors gave final approval for publication and agreed to be held accountable for the work performed therein.

Competing interests. No financial and non-financial competing interests were declared on behalf of all authors.

Funding. This study was supported by the SPP 1529 priority pro-gramme 'Adaptomics' grants of the German Research Foundation (DFG) to K.T. and M.G. (grant nos. TI 338/10-1 and TI 338/10-2) and to S.C. (grant no. CL 152/9-2).

Acknowledgements. We are grateful to Ute Krämer and Ricardo Stein for introducing us to the topic of metal-hyperaccumulating plants, and to Mira Hoch, Bettina Springer and Anne Rysavy for the collection of *A. halleri* and soil in the field, the preparation of contaminated soil, and propagation of *A. halleri* in the greenhouse.

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
