## [Peer Review File · Proceedings of the Royal Society B: Biological Sciences]

Review History

RSPB-2020-3037.R0 (Original submission)

Review form: Reviewer 1

Recommendation

Accept with minor revision (please list in comments)

Scientific importance: Is the manuscript an original and important contribution to its field?

Good

General interest: Is the paper of sufficient general interest?

Good

Quality of the paper: Is the overall quality of the paper suitable?

Good

Is the length of the paper justified?

Yes

Should the paper be seen by a specialist statistical reviewer?

No

Do you have any concerns about statistical analyses in this paper? If so, please specify them explicitly in your report.

No

It is a condition of publication that authors make their supporting data, code and materials available - either as supplementary material or hosted in an external repository. Please rate, if applicable, the supporting data on the following criteria.

Is it accessible?

Yes

Is it clear?

Yes

Is it adequate?

Yes

Do you have any ethical concerns with this paper?

No

Comments to the Author

The authors present an original study regarding foraging and sharing of Cd in response to simulated herbivory. This study is in line with ecological studies based on remarkable (hyper)-accumulating plant species. These plants are often studied in the context of polluted sites remediation. They also offer great opportunities to address important ecological questions, as demonstrated once again by the authors. Overall, I found the ecological questions addressed in this study important, and properly addressed by a smart and elegant experimental design, and interpretation of the results are convincing. If I have some questions/remarks (see above), I'm very positive regarding this study.

First experiment:

- the functional interpretation of the modification of roots biomass proportions in different pots for non-metalliferous plants would have been even more convincing if the authors had provided the proof of the increase of Cd concentration in corresponding plant leaves. Why not providing these results if available? (see line 137: However, shoot biomass of six plants per treatment could not be analyzed, as they were used for additional chemical analysis (data not shown). This point is not mandatory since the second experiment provides some evidences of higher Cd accumulation induced by simulated herbivory (but it does not provide results of Cd accumulation in relation with root foraging).

- The authors insist on the higher root allocation to Cd rich pots in case of simulated herbivory for non-metalliferous plants (line 196: i.e. more roots were allocated to Cd rich soil when the plants perceived herbivory). For these plants, because root biomass in low Cd pots decrease with herbivory, it is true that relatively more root biomass is found in high Cd pots in case of herbivory. But root biomass in high Cd pots does not change with herbivory. We might better say that root allocation in high Cd pots is maintained while it decreases in Low Cd pots. Functionally, the implications are the same: a guaranteed access to Cd, to improve plant defense.

Second experiment:

- If results of shoot biomass are presented (Fig 4A), I did not find a clear interpretation of these results in the discussion. Moreover, it is difficult to compare shoot allocation between

different ramets if we don't have the root biomass results. For instance, the authors indicate for non-metalliferous plants that "under control conditions they allocated more shoot biomass to ramets in the low-Cd pots – line 203). But if root biomasses of the same ramets are also higher, shoot mass fractions would remain the same, and it is difficult to conclude in terms of "allocation". No available results regarding root biomasses of the different ramets in this experiment? In addition, "allocation" should be used with care because mass fraction of different plant organs can vary with plant size, and plant size is impacted by soil contamination.

- How do the authors interpret the increase in shoot biomass in high-Cd pots only when simulated herbivory concerned ramets in low cd pots? The two herbivory treatments led to high Cd sharing between ramets for non-metalliferous plants. But it led to different results regarding shoot growth. I did not find the explanation of this result.

Methods:

- Two different soils were used in the two experiments. Is it problematic? Any limitation for this study? (a brief justification in the methods would be welcome).

- Line 156: "in half of the pairs the mother ramets were assigned to the high-Cd pot, while in the other half the daughter ramets were." ? assigned? (Missing word). Any interest to consider this (mother or daughter ramets) during analysis?

- Line 174: "while root biomass was analysed using a gamma probability distribution with an identity link function". Is it better/more accurate than a simpler/more standard log-transformation of the data?

- Please use letter labelling in the figures (a and b for pairwise multiple comparisons) constantly (letter a for lowest values, letter b for highest values).

Terminology:

- The authors used "metalliferous soils – non-metalliferous soils" to present results according to plant origin (metalliferous or not). It is sometimes confusing with the High and Low Cd soil contamination in pots. Possible to use mettaliferous origin or mettaliferous plant instead?

- The "local" and "remote" terms for the second experiment are not straightforward. Possible to be more explicit for the ease of reading? (for instance: herbivory in High Cd pots and herbivory in Low Cd pots)

- In the text, Cd is sometimes referred to as a 'resource'. (line 218: "Our study provides support for the idea that foraging and resource uptake in plants can be induced by herbivory"). I understand that Cd is a 'resource' for plant defense. But the term 'resource' is more often used for resources involved in primary metabolism in the literature, isn't it? (very minor point, it is more a question than a recommendation).

Review form: Reviewer 2

Recommendation

Major revision is needed (please make suggestions in comments)

Scientific importance: Is the manuscript an original and important contribution to its field?

Excellent

General interest: Is the paper of sufficient general interest?

Excellent

Quality of the paper: Is the overall quality of the paper suitable?

Good

Is the length of the paper justified?

Yes

Should the paper be seen by a specialist statistical reviewer?

Yes

Do you have any concerns about statistical analyses in this paper? If so, please specify them explicitly in your report.

No

It is a condition of publication that authors make their supporting data, code and materials available - either as supplementary material or hosted in an external repository. Please rate, if applicable, the supporting data on the following criteria.

Is it accessible?

Yes

Is it clear?

Yes

Is it adequate?

Yes

Do you have any ethical concerns with this paper?

No

Comments to the Author

Major comments:

MJ1 * Soil from two different sites was used in Experiment 1 (root foraging experiment). For contextualising the results and allowing for comparability between studies additional soil parameters would be helpful, as they are known modulators of root growth: soil type, bulk density, pH, EC. Please provide the requested information or comment on why you think that Cd is the only factor involved in affecting root growth under herbivory in the low and high Cd treatment.

MJ2* The outcome of Experiment 2 is very interesting. Enhanced Cd uptake and transfer to shoots and between ramets in response to herbivory (local and remote, in plants of non-metalliferous origin). However, after reading title and abstract, the reader would expect root biomass data in addition to shoot biomass and shoot Cd concentrations. As presented currently the "clonal foraging" experiment is a "clonal Cd uptake and transfer experiment". Please adjust the manuscript accordingly or explain your definition of root foraging in the introduction. According to my understanding "root foraging" describes root placement patterns rather than root physiological responses.

MJ3* In experiment 2 you propose that "increased Cd sharing" in response to herbivory occurs in plants originating from metalliferous sites. There is a tendency, but I do not see a significant increase in Cd concentrations in low-Cd ramets nor a significant decrease in high-Cd ramets relative to the respective controls. Would there be other variables or derived variables that could provide evidence for increased Cd sharing or transfer?

Minor comments per section:

Abstract:

* l6: include foraging "for Cd", to be more specific

* Which result indicates that the simulated herbivory applied in your study was "more-detrimental" than enhanced Cd uptake?* I9-I10: "Moreover, these responses were exhibited particularly by plants from non-metalliferous soils, which have lower metal tolerance." This statement implies that the directionality of change was similar in plants from the two different origins. Could you be more precise about the distinct results in the two groups already in this section of the manuscript?

Introduction

I38 please also include information on the effect of leaf herbivory on root morphology or physiology (cf. Kaplan, I., Halitschke, R., Kessler, A., Sardanelli, S., & Denno, R. F. (2008). Constitutive and induced defenses to herbivory in above-and belowground plant tissues. *Ecology*, 89(2), 392-406., or more recent literature).

Materials and Methods

* please include information on the watering regime used in both experiments

* please specify the instrumentation used for ICP-OES

Results

* I196-197: This statement is ambivalent. A higher "proportion of the total root biomass" was allocated to Cd rich soil, while the biomass in Cd depleted soil was lower when compared to control plants.

* I198 cf. (MJ2) consistent in terms of foraging?

* Fig 4: in the method section you state that leaves from 6 plants per treatment were analysed for their Cd content. Please explain the higher number of replicates indicated in Fig 4B.

* the term "induction" (local and remote) describes the result of a treatment. As your results also show no significant induction relative to untreated controls (in the Local Herbivory treatment on plants originating from metalliferous soils) it could be worthwhile to consider the use of "local and remote herbivory" to describe your treatment, instead of "induction" in Figure 4 and the manuscript text where appropriate.

Discussion

*I238. "shown mainly in ramets originating from non-metalliferous soils" My interpretation of the data shown is that enhanced foraging (but cf MJ1) and Cd uptake in response to herbivory was apparent in ramets from non-metalliferous soils and that plants from metalliferous soils show a tendency of increased Cd transfer between ramets. Please comment.*Do your results indicate that the Cd treatment employed is toxic to *A. halleri* originating from metalliferous soils? Are toxicity levels of *A. halleri* of different origins published in the literature?

Decision letter (RSPB-2020-3037.R0)

28-Jan-2021

Dear Dr Mohiley:

I am writing to inform you that your manuscript RSPB-2020-3037 entitled "Between the devil and the deep blue sea: herbivory induces foraging for poison in plants" has, in its current form, been rejected for publication in *Proceedings B*.

This action has been taken on the advice of referees, who have recommended that substantial revisions are necessary. With this in mind we would be happy to consider a resubmission,

provided the comments of the referees are fully addressed. However please note that this is not a provisional acceptance.

Sincerely,
 Professor Gary Carvalho
 mailto: proceedingsb@royalsociety.org

Associate Editor
 Board Member: 1
 Comments to Author:

This is an interesting study which tested the hypothesis that foraging for and sharing of heavy metals can be induced by herbivory. To achieve this the authors tested foraging responses for heavy metals, specifically Cadmium in a metal hyperaccumulating clonal plant *Arabidopsis halleri* and posed the question, whether simulated damage induced both increased Cd uptake and increased Cd sharing between *A. halleri* ramets. Key attributes of this work are that the study addresses a knowledge gap in the role of herbivory in inducing and modulating foraging decisions in plants; use of simulated herbivory by mechanical damage with jasmonic acid application plus appropriate controls following standard methods and sound clonal foraging experiments backed by appropriate statistical analysis. However, there are a few technical weaknesses in the study, as rightly pointed out by the two reviewers including use of two different soils in the two different experiments- would soil type not affect the output of these experiments including shoot and root biomass and Cd allocation? Another point of concern is whether other variables in the soil may impact on Cd allocation? Perhaps the authors should consider discussing limitations of the study.

Reviewer(s)' Comments to Author:
 Referee: 1

Comments to the Author(s)

The authors present an original study regarding foraging and sharing of Cd in response to simulated herbivory. This study is in line with ecological studies based on remarkable (hyper)-accumulating plant species. These plants are often studied in the context of polluted sites remediation. They also offer great opportunities to address important ecological questions, as

demonstrated once again by the authors. Overall, I found the ecological questions addressed in this study important, and properly addressed by a smart and elegant experimental design, and interpretation of the results are convincing. If I have some questions/remarks (see above), I'm very positive regarding this study.

First experiment:

- the functional interpretation of the modification of roots biomass proportions in different pots for non-metalliferous plants would have been even more convincing if the authors had provided the proof of the increase of Cd concentration in corresponding plant leaves. Why not providing these results if available? (see line 137: However, shoot biomass of six plants per treatment could not be analyzed, as they were used for additional chemical analysis (data not shown). This point is not mandatory since the second experiment provides some evidences of higher Cd accumulation induced by simulated herbivory (but it does not provide results of Cd accumulation in relation with root foraging).

- The authors insist on the higher root allocation to Cd rich pots in case of simulated herbivory for non-metalliferous plants (line 196: i.e. more roots were allocated to Cd rich soil when the plants perceived herbivory). For these plants, because root biomass in low Cd pots decrease with herbivory, it is true that relatively more root biomass is found in high Cd pots in case of herbivory. But root biomass in high Cd pots does not change with herbivory. We might better say that root allocation in high Cd pots is maintained while it decreases in Low Cd pots. Functionally, the implications are the same: a guaranteed access to Cd, to improve plant defense.

Second experiment:

- If results of shoot biomass are presented (Fig 4A), I did not find a clear interpretation of these results in the discussion. Moreover, it is difficult to compare shoot allocation between different ramets if we don't have the root biomass results. For instance, the authors indicate for non-metalliferous plants that "under control conditions they allocated more shoot biomass to ramets in the low-Cd pots – line 203). But if root biomasses of the same ramets are also higher, shoot mass fractions would remain the same, and it is difficult to conclude in terms of "allocation". No available results regarding root biomasses of the different ramets in this experiment? In addition, "allocation" should be used with care because mass fraction of different plant organs can vary with plant size, and plant size is impacted by soil contamination.

- How do the authors interpret the increase in shoot biomass in high-Cd pots only when simulated herbivory concerned ramets in low cd pots? The two herbivory treatments led to high Cd sharing between ramets for non-metalliferous plants. But it led to different results regarding shoot growth. I did not find the explanation of this result.

Methods:

- Two different soils were used in the two experiments. Is it problematic? Any limitation for this study? (a brief justification in the methods would be welcome).

- Line 156: "in half of the pairs the mother ramets were assigned to the high-Cd pot, while in the other half the daughter ramets were." ? assigned? (Missing word). Any interest to consider this (mother or daughter ramets) during analysis?

- Line 174: "while root biomass was analysed using a gamma probability distribution with an identity link function". Is it better/more accurate than a simpler/more standard log-transformation of the data?

- Please use letter labelling in the figures (a and b for pairwise multiple comparisons) constantly (letter a for lowest values, letter b for highest values).

Terminology:

- The authors used “metalliferous soils – non-metalliferous soils” to present results according to plant origin (metalliferous or not). It is sometimes confusing with the High and Low Cd soil contamination in pots. Possible to use metalliferous origin or metalliferous plant instead?
- The “local” and “remote” terms for the second experiment are not straightforward. Possible to be more explicit for the ease of reading? (for instance: herbivory in High Cd pots and herbivory in Low Cd pots)
- In the text, Cd is sometimes referred to as a ‘resource’. (line 218: “Our study provides support for the idea that foraging and resource uptake in plants can be induced by herbivory”). I understand that Cd is a ‘resource’ for plant defense. But the term ‘resource’ is more often used for resources involved in primary metabolism in the literature, isn’t it? (very minor point, it is more a question than a recommendation).

Referee: 2

Comments to the Author(s)

Major comments:

MJ1 * Soil from two different sites was used in Experiment 1 (root foraging experiment). For contextualising the results and allowing for comparability between studies additional soil parameters would be helpful, as they are known modulators of root growth: soil type, bulk density, pH, EC. Please provide the requested information or comment on why you think that Cd is the only factor involved in affecting root growth under herbivory in the low and high Cd treatment.

MJ2* The outcome of Experiment 2 is very interesting. Enhanced Cd uptake and transfer to shoots and between ramets in response to herbivory (local and remote, in plants of non-metalliferous origin). However, after reading title and abstract, the reader would expect root biomass data in addition to shoot biomass and shoot Cd concentrations. As presented currently the "clonal foraging" experiment is a "clonal Cd uptake and transfer experiment". Please adjust the manuscript accordingly or explain your definition of root foraging in the introduction. According to my understanding "root foraging" describes root placement patterns rather than root physiological responses.

MJ3* In experiment 2 you propose that "increased Cd sharing" in response to herbivory occurs in plants originating from metalliferous sites. There is a tendency, but I do not see a significant increase in Cd concentrations in low-Cd ramets nor a significant decrease in high-Cd ramets relative to the respective controls. Would there be other variables or derived variables that could provide evidence for increased Cd sharing or transfer?

Minor comments per section:

Abstract:

* l6: include foraging "for Cd", to be more specific

* Which result indicates that the simulated herbivory applied in your study was "more-detrimental" than enhanced Cd uptake?* l9-l10: "Moreover, these responses were exhibited particularly by plants from non-metalliferous soils, which have lower metal tolerance." This statement implies that the directionality of change was similar in plants from the two different origins. Could you be more precise about the distinct results in the two groups already in this section of the manuscript?

Introduction

l38 please also include information on the effect of leaf herbivory on root morphology or physiology (cf. Kaplan, I., Halitschke, R., Kessler, A., Sardanelli, S., & Denno, R. F. (2008).

Constitutive and induced defenses to herbivory in above-and belowground plant tissues. Ecology, 89(2), 392-406., or more recent literature).

Materials and Methods

- * please include information on the watering regime used in both experiments
- * please specify the instrumentation used for ICP-OES

Results

- * l196-197: This statement is ambivalent. A higher "proportion of the total root biomass" was allocated to Cd rich soil, while the biomass in Cd depleted soil was lower when compared to control plants.
- * l198 cf. (MJ2) consistent in terms of foraging?
- * Fig 4: in the method section you state that leaves from 6 plants per treatment were analysed for their Cd content. Please explain the higher number of replicates indicated in Fig 4B.
- * the term "induction" (local and remote) describes the result of a treatment. As your results also show no significant induction relative to untreated controls (in the Local Herbivory treatment on plants originating from metalliferous soils) it could be worthwhile to consider the use of "local and remote herbivory" to describe your treatment, instead of "induction" in Figure 4 and the manuscript text where appropriate.

Discussion

- *l238. "shown mainly in ramets originating from non-metalliferous soils" My interpretation of the data shown is that enhanced foraging (but cf MJ1) and Cd uptake in response to herbivory was apparent in ramets from non-metalliferous soils and that plants from metalliferous soils show a tendency of increased Cd transfer between ramets. Please comment.*Do your results indicate that the Cd treatment employed is toxic to A. halleri originating from metalliferous soils? Are toxicity levels of A. halleri of different origins published in the literature?

Author's Response to Decision Letter for (RSPB-2020-3037.R0)

See Appendix A.

RSPB-2021-1682.R0

Review form: Reviewer 1

Recommendation

Accept as is

Scientific importance: Is the manuscript an original and important contribution to its field?

Good

General interest: Is the paper of sufficient general interest?

Good

Quality of the paper: Is the overall quality of the paper suitable?

Good

Is the length of the paper justified?

Yes

Should the paper be seen by a specialist statistical reviewer?

No

Do you have any concerns about statistical analyses in this paper? If so, please specify them explicitly in your report.

No

It is a condition of publication that authors make their supporting data, code and materials available - either as supplementary material or hosted in an external repository. Please rate, if applicable, the supporting data on the following criteria.

Is it accessible?

Yes

Is it clear?

Yes

Is it adequate?

Yes

Do you have any ethical concerns with this paper?

No

Comments to the Author

I thank the authors for their responses and modifications.

I have no other remarks.

I found the article convincing overall and I think it now deserves publication.

Review form: Reviewer 2

Recommendation

Accept as is

Scientific importance: Is the manuscript an original and important contribution to its field?

Good

General interest: Is the paper of sufficient general interest?

Good

Quality of the paper: Is the overall quality of the paper suitable?

Good

Is the length of the paper justified?

Yes

Should the paper be seen by a specialist statistical reviewer?

No

Do you have any concerns about statistical analyses in this paper? If so, please specify them explicitly in your report.

No

It is a condition of publication that authors make their supporting data, code and materials available - either as supplementary material or hosted in an external repository. Please rate, if applicable, the supporting data on the following criteria.

Is it accessible?

Yes

Is it clear?

Yes

Is it adequate?

Yes

Do you have any ethical concerns with this paper?

No

Comments to the Author

My previous comments have been convincingly addressed by the authors. This is a sound study with high ecological relevance.

I really would just suggest three minor changes:

Title/keywords: mentioning the term "cadmium" in the title or in the keyword section might be beneficial for reader accessibility

Methods:

l120 which product was used from Bischoff GmbH, soil or contaminant? Please specify in the text.

l260 typo: lack of statistically significant effect

Decision letter (RSPB-2021-1682.R0)

31-Aug-2021

Dear Dr Gruntman

I am pleased to inform you that your manuscript RSPB-2021-1682 entitled "Between the devil and the deep blue sea: herbivory induces foraging for and uptake of poison in plants" has been accepted for publication in Proceedings B.

The referee(s) have recommended publication, but also suggest some minor revisions to your manuscript. Therefore, I invite you to respond to the referee(s)' comments and revise your manuscript. Because the schedule for publication is very tight, it is a condition of publication that you submit the revised version of your manuscript within 7 days. If you do not think you will be able to meet this date please let us know.

When submitting your revised manuscript, you will be able to respond to the comments made by the referee(s) and upload a file "Response to Referees". You can use this to document any changes you make to the original manuscript. We require a copy of the manuscript with revisions made

since the previous version marked as 'tracked changes' to be included in the 'response to referees' document.

Sincerely,
Professor Gary Carvalho
mailto: proceedingsb@royalsociety.org

Associate Editor

Comments to Author:

The content of the manuscript is in good shape. However, there are two minor issues that must be addressed, both raised by Review 2: First, edits to the title to make it more specific to fit the study; second, clarification in the methods.

Reviewer(s)' Comments to Author:

Referee: 2

Comments to the Author(s).

My previous comments have been convincingly addressed by the authors. This is a sound study with high ecological relevance.

I really would just suggest three minor changes:

Title/keywords: mentioning the term "cadmium" in the title or in the keyword section might be beneficial for reader accessibility

Methods:

l120 which product was used from Bischoff GmbH, soil or contaminant? Please specify in the text.

l260 typo: lack of statistically significant effect

Referee: 1

Comments to the Author(s).

I thank the authors for their responses and modifications.

I have no other remarks.

I found the article convincing overall and I think it now deserves publication.

Author's Response to Decision Letter for (RSPB-2021-1682.R0)

See Appendix B.

Decision letter (RSPB-2021-1682.R1)

07-Sep-2021

Dear Dr Gruntman

I am pleased to inform you that your manuscript entitled "Between the devil and the deep blue sea: herbivory induces foraging for and uptake of cadmium in a metal hyperaccumulating plant" has been accepted for publication in Proceedings B.

Your article has been estimated as being 9 pages long. Our Production Office will be able to confirm the exact length at proof stage.

Data Accessibility section

Open Access

Paper charges

Sincerely,

Appendix A

Dear Editor,

Enclosed please find our revised manuscript (RSPB-2020-3037) entitled "Between the devil and the deep blue sea: herbivory induces foraging for poison in plants". We wish to thank both you and the reviewers for the very helpful comments and suggestions, which have enabled us to greatly improve the quality and clarity of the manuscript and for providing the possibility to submit this revised version.

The manuscript has been revised according to the reviewers' suggestions and we hope that our revision will be found suitable for publication in Proceedings of the Royal Society B. The following is an itemized discussion of our response to the points raised by the reviewers.

Associate Editor

This is an interesting study which tested the hypothesis that foraging for and sharing of heavy metals can be induced by herbivory. To achieve this the authors tested foraging responses for heavy metals, specifically Cadmium in a metal hyperaccumulating clonal plant *Arabidopsis halleri* and posed the question, whether simulated damage induced both increased Cd uptake and increased Cd sharing between *A. halleri* ramets. Key attributes of this work are that the study addresses a knowledge gap in the role of herbivory in inducing and modulating foraging decisions in plants; use of simulated herbivory by mechanical damage with jasmonic acid application plus appropriate controls following standard methods and sound clonal foraging experiments backed by appropriate statistical analysis. However, there are a few technical weaknesses in the study, as rightly pointed out by the two reviewers including use of two different soils in the two different experiments- would soil type not affect the output of these experiments including shoot and root biomass and Cd allocation? Another point of concern is whether other variables in the soil may impact on Cd allocation? Perhaps the authors should consider discussing limitations of the study.

We thank the Editor for this positive assessment and suggestions.

As we replied to the concern raised by Reviewer 1, we agree that artificially contaminated soil might differ from field contaminated soil in various properties, including Cd availability. However, we do not believe that these variations between experiments could have affected the different responses to simulated herbivory, which were observed within each experiment. Nevertheless, as suggested by the Editor, we have now added a paragraph on this potential limitation of the study in the discussion section (lines 276-280), as well as a justification for our use of different soils in the methods section (lines 110-112).

In addition, Reviewer 2 raised a concern regarding differences between the soils collected in the metalliferous vs. non-metalliferous sites, in other parameters than Cd availability. In

response to this concern, we added pH values for the two soils line (108-109), as well as explained the methodologies we used to minimize potential differences between the soils in their physical properties, nutrient availability or the presence of mutualist or antagonist microorganisms. These methodologies included sieving the soils, adding a slow-release fertiliser and sterilizing the soils, as is now clarified in the methods section (Lines 102-109)

Referee: 1

The authors present an original study regarding foraging and sharing of Cd in response to simulated herbivory. This study is in line with ecological studies based on remarkable (hyper)-accumulating plant species. These plants are often studied in the context of polluted sites remediation. They also offer great opportunities to address important ecological questions, as demonstrated once again by the authors. Overall, I found the ecological questions addressed in this study important, and properly addressed by a smart and elegant experimental design, and interpretation of the results are convincing. If I have some questions/remarks (see above), I'm very positive regarding this study.

We thank the reviewer for the positive assessment of the manuscript.

First experiment:

- the functional interpretation of the modification of roots biomass proportions in different pots for non-metalliferous plants would have been even more convincing if the authors had provided the proof of the increase of Cd concentration in corresponding plant leaves. Why not providing these results if available? (see line 137: However, shoot biomass of six plants per treatment could not be analyzed, as they were used for additional chemical analysis (data not shown). This point is not mandatory since the second experiment provides some evidences of higher Cd accumulation induced by simulated herbivory (but it does not provide results of Cd accumulation in relation with root foraging).

We agree with the reviewer, but unfortunately, we do not have the data on the amount of Cd accumulated in the first experiment, as the leaf material was used to analyse glucosinolate concentrations as a defence response as part of a different study.

- The authors insist on the higher root allocation to Cd rich pots in case of simulated herbivory for non-metalliferous plants (line 196: i.e. more roots were allocated to Cd rich soil when the plants perceived herbivory). For these plants, because root biomass in low Cd pots decrease with herbivory, it is true that relatively more root biomass is found in high Cd pots in case of herbivory. But root biomass in high Cd pots does not change with herbivory. We might better say that root allocation in high Cd pots is maintained while it decreases in Low

Cd pots. Functionally, the implications are the same: a guaranteed access to Cd, to improve plant defence.

We agree with the reviewer and have accordingly changed this interpretation of the results (lines 208-209 in the results and 234-236 in the discussion).

Second experiment:

- If results of shoot biomass are presented (Fig 4A), I did not find a clear interpretation of these results in the discussion. Moreover, it is difficult to compare shoot allocation between different ramets if we don't have the root biomass results. For instance, the authors indicate for non-metalliferous plants that "under control conditions they allocated more shoot biomass to ramets in the low-Cd pots – line 203). But if root biomasses of the same ramets are also higher, shoot mass fractions would remain the same, and it is difficult to conclude in terms of "allocation". No available results regarding root biomasses of the different ramets in this experiment? In addition, "allocation" should be used with care because mass fraction of different plant organs can vary with plant size, and plant size is impacted by soil contamination.

We agree with the reviewer that belowground biomass would have provided valuable information here. However, in this experiment we encountered technical difficulties in root washing due to the use of local soil that was rich in clay and separating the roots unfortunately proved impossible. We also agree that our use of the term "allocation" might be misleading in this case, because we used it here to refer to aboveground biomass allocation between connected ramets rather than shoot/root allocation. We have therefore revised this section and now refrain from using the term.

- How do the authors interpret the increase in shoot biomass in high-Cd pots only when simulated herbivory concerned ramets in low cd pots? The two herbivory treatments led to high Cd sharing between ramets for non-metalliferous plants. But it led to different results regarding shoot growth. I did not find the explanation of this result.

We thank reviewer for the comment. The aboveground biomass results suggest that plants from non-metalliferous origin might incur a cost of Cd accumulation and thus mostly avoided growing in the high-Cd pots. However, we agree that this hypothesis does not provide an explanation for the increase in biomass in the high-Cd pot when herbivory was simulated in the low-Cd one. It might be possible that simulated herbivory had a negative effect on ramet biomass in the low-Cd pot, which is compensated for by a biomass increase in the high-Cd one. However, we feel that discussing this speculation (and more generally the results regarding ramet biomass) is less related to the main scope of this study and therefore preferred not to refer to it in the discussion section.

Methods:

- Two different soils were used in the two experiments. Is it problematic? Any limitation for this study? (a brief justification in the methods would be welcome).

We thank the reviewer for the concern. We used two different soils in the experiment as unfortunately we could not obtain additional field soil for the second experiment. We therefore chose to use similar local field soil, which we artificially contaminated, rather than potting soil, which could have been very different. We added this justification to the methods as suggested by the reviewer (lines 110-112). However, although we agree that artificially contaminated soil might differ from field contaminated soil in various properties, we do not believe that these differences could have affected the different responses to simulated herbivory, which were observed within each experiment. Nevertheless, we have now added a discussion on this potential limitation of the study (lines 276-281).

- Line 156: "in half of the pairs the mother ramets were assigned to the high-Cd pot, while in the other half the daughter ramets were." ? assigned? (Missing word). Any interest to consider this (mother or daughter ramets) during analysis?

We thank the reviewer for this comment and have accordingly added the missing "assigned to it" to the text. In addition, we did consider the mother vs., daughter ramets in our initial analysis to learn if it might affect Cd allocation patterns but removed it due to lower model fit (a higher AIC value) and lack of statically-significant effect. This explanation has been added to the methods (lines 191-193).

- Line 174: "while root biomass was analysed using a gamma probability distribution with an identity link function". Is it better/more accurate than a simpler/more standard log-transformation of the data?

A gamma probability distribution was used here as it resulted in the best model fit (with the lowest AIC value).

- Please use letter labelling in the figures (a and b for pairwise multiple comparisons) constantly (letter a for lowest values, letter b for highest values).

We thank the reviewer for the comment. We agree with the reviewer and have made the suggested change.

Terminology:

- The authors used “metalliferous soils – non-metalliferous soils” to present results according to plant origin (metalliferous or not). It is sometimes confusing with the High and Low Cd soil contamination in pots. Possible to use metalliferous origin or metalliferous plant instead?

We agree with the reviewer and have thus changed the term "soils to "origin" when referring to plant origin throughout the manuscript.

- The “local” and “remote” terms for the second experiment are not straightforward. Possible to be more explicit for the ease of reading? (for instance: herbivory in High Cd pots and herbivory in Low Cd pots)

We thank the reviewer for this comment and changed these terms per the reviewer's suggestion.

- In the text, Cd is sometimes referred to as a ‘resource’. (line 218: “Our study provides support for the idea that foraging and resource uptake in plants can be induced by herbivory”). I understand that Cd is a ‘resource’ for plant defense. But the term ‘resource’ is more often used for resources involved in primary metabolism in the literature, isn't it? (very minor point, it is more a question than a recommendation).

We thank the reviewer for the concern. We agree that term “resource” for plant defence has been used in regard to primary metabolism, as is explained in the introduction (lines 40-42). However, in the case of metal hyperaccumulating plants such as *A. halleri* we suggest that heavy metals might also be a resource used for herbivore defence (lines 52-56).

Referee: 2

Major comments:

MJ1 * Soil from two different sites was used in Experiment 1 (root foraging experiment). For contextualising the results and allowing for comparability between studies additional soil parameters would be helpful, as they are known modulators of root growth: soil type, bulk density, pH , EC. Please provide the requested information or comment on why you think that Cd is the only factor involved in affecting root growth under herbivory in the low and high Cd treatment.

We thank the reviewer for the concern. Unfortunately, we do not have additional parameters except for pH, which was 5.4 and 5.8 metalliferous and non-metalliferous soil mixtures, respectively, as is now added to the text (line 108-109). However, to minimize potential differences between the soils that might stem from differences in their physical properties or nutrient availability, the soils were sieved through a 2mm mesh size and added with 10 g of slow-release fertiliser. Moreover, in order to minimize potential differences between the soils

due to mutualist or antagonist microorganisms the soils were sterilized. We have now added this clarification in the text (lines 100-109).

MJ2* The outcome of Experiment 2 is very interesting. Enhanced Cd uptake and transfer to shoots and between ramets in response to herbivory (local and remote, in plants of non-metalliferous origin). However, after reading title and abstract, the reader would expect root biomass data in addition to shoot biomass and shoot Cd concentrations. As presented currently the "clonal foraging" experiment is a "clonal Cd uptake and transfer experiment". Please adjust the manuscript accordingly or explain your definition of root foraging in the introduction. According to my understanding "root foraging" describes root placement patterns rather than root physiological responses.

We thank the reviewer for the comment and have accordingly changed the definition of this experiment to a "clonal-sharing experiment", and corrected the title, abstract and introduction of the manuscript.

MJ3* In experiment 2 you propose that "increased Cd sharing" in response to herbivory occurs in plants originating from metalliferous sites. There is a tendency, but I do not see a significant increase in Cd concentrations in low-Cd ramets nor a significant decrease in high-Cd ramets relative to the respective controls. Would there be other variables or derived variables that could provide evidence for increased Cd sharing or transfer?

Here, we suggest that the extent of Cd sharing can be estimated as the differences in Cd accumulation between ramets in the high vs. low-Cd pots within treatments. Thus, plants from metalliferous origin had significantly greater Cd in ramets in the high-Cd pot under control conditions or when the herbivory was simulated in the high-Cd pot, suggesting low sharing between ramets. However, when herbivory was simulated in the low-Cd pot – these plants exhibited similar Cd levels in both ramets, indicating Cd sharing between ramets.

Minor comments per section:

Abstract:

* 16: include foraging "for Cd", to be more specific

Revised as suggested.

* Which result indicates that the simulated herbivory applied in your study was "more-detrimental" than enhanced Cd uptake?* 19-110: "Moreover, these responses were exhibited particularly by plants from non-metalliferous soils, which have lower metal tolerance." This statement implies that the directionality of change was similar in plants from the two different

origins. Could you be more precise about the distinct results in the two groups already in this section of the manuscript?

We thank the reviewer for this comment and have accordingly removed the term "more-detrimental" from the abstract. Moreover, in light of the comment we have revised the description of the results in the abstract to be more precise about the results.

Introduction

138 please also include information on the effect of leaf herbivory on root morphology or physiology (cf. Kaplan, I., Halitschke, R., Kessler, A., Sardanelli, S., & Denno, R. F. (2008). Constitutive and induced defenses to herbivory in above-and belowground plant tissues. *Ecology*, 89(2), 392-406., or more recent literature).

Per the reviewer's suggestion, a reference to this highly relevant study was added here. However, we think that specifying the particular effects of leaf herbivory on the secondary metabolites produced in roots (the study did not examine effects on root morphology or physiology) deviated from the subject of this paragraph, and therefore preferred not to elaborate on the subject here.

Materials and Methods

* please include information on the watering regime used in both experiments

According to this comment, information on watering regimes was included for both experiments (138-139).

* please specify the instrumentation used for ICP-OES

The instrument was specified as suggested (line 173).

Results

* 1196-197: This statement is ambivalent. A higher "proportion of the total root biomass" was allocated to Cd rich soil, while the biomass in Cd depleted soil was lower when compared to control plants.

According to the reviewer's comment, we have changed the sentence to clarify this result.

* 1198 cf. (MJ2) consistent in terms of foraging?

As suggested in our response to MJ2, we have changed the definition of this experiment to a "clonal-sharing experiment".

* Fig 4: in the method section you state that leaves from 6 plants per treatment were analysed for their Cd content. Please explain the higher number of replicates indicated in Fig 4B.

In the methods section we stated that in the first experiment (Fig. 3) "shoot biomass of six plants per treatment could not be analysed, as they were used for additional chemical analysis". We therefore had a lower number of replicates for shoot biomass in this experiment. However, in the second experiment, all the plants were sent for Cd content analysis.

* the term "induction" (local and remote) describes the result of a treatment. As your results also show no significant induction relative to untreated controls (in the Local Herbivory treatment on plants originating from metalliferous soils) it could be worthwhile to consider the use of "local and remote herbivory" to describe your treatment, instead of "induction" in Figure 4 and the manuscript text where appropriate.

We thank the reviewer for this comment, and in accordance also with a similar comment from reviewer 1, we have now changed the terms to "simulated herbivory on high-Cd or low-Cd pots".

Discussion

*1238. "shown mainly in ramets originating from non-metalliferous soils" My interpretation of the data shown is that enhanced foraging (but cf MJ1) and Cd uptake in response to herbivory was apparent in ramets from non-metalliferous soils and that plants from metalliferous soils show a tendency of increased Cd transfer between ramets. Please comment.

We agree with the reviewer that Cd sharing increased in plants from metalliferous soils and have commented on this result in the discussion section in the previous version of this manuscript (lines 271-275).

*Do your results indicate that the Cd treatment employed is toxic to *A. halleri* originating from metalliferous soils? Are toxicity levels of *A. halleri* of different origins published in the literature?

In this study we did not examine potential toxic effects of Cd on plants from either metalliferous or non-metalliferous soils. However, as mentioned later in the discussion (lines 251-266), previous studies (including our own with the same genotypes) have shown that *A. halleri* from non-metalliferous populations are less tolerant to Cd compared to plants from metalliferous soils (Gruntman *et al.*, 2016).

Appendix B

Dear Editor,

Enclosed please find our revised manuscript (RSPB- 2021-1682). We wish to thank again both you and the reviewers for the very helpful reviews and for finding this manuscript suitable for publication in Proceedings of the Royal Society B. The manuscript was revised according to the following suggestions raised by referee 1:

Title/keywords: mentioning the term "cadmium" in the title or in the keyword section might be beneficial for reader accessibility

As suggested by the reviewer, the term "cadmium" was added to the title instead of "poison". Moreover, to make the title more specific to the study, as suggested by the Editor, we changed "plants" to "a metal hyperaccumulating plant".

L120 which product was used from Bischoff GmbH, soil or contaminant? Please specify in the text.

The sentence was revised to clarify this.

L260 typo: lack of statistically significant effect

Corrected.